# Domain Adversarial Transfer Learning Bearing Fault Diagnosis Model Incorporating Structural Adjustment Modules

**DOI:** 10.3390/s25061851

**Published:** 2025-03-17

**Authors:** Zhidan Zhong, Hao Xie, Zhenxin Wang, Zhihui Zhang

**Affiliations:** School of Mechanical and Electrical Engineering, Henan University of Science and Technology, Luoyang 471023, China; 221401150725@stu.haust.edu.cn (H.X.); xzw9436946@163.com (Z.W.); zzhui2022@163.com (Z.Z.)

**Keywords:** bearing failure, domain adversarial, transfer learning, Optuna

## Abstract

With the improvement in industrial equipment intelligence and reliability requirements, bearing fault diagnosis has become a key technology to ensure the stable operation of mechanical equipment. Traditional bearing fault diagnosis methods are ineffective in diagnosing complex faults and mostly rely on the manual adjustment of hyperparameters. To this end, this paper proposes a domain adversarial migratory learning bearing fault diagnosis model incorporating structural adjustment modules. First, the pre-trained model of the source domain is applied to the target domain dataset through an adversarial domain adaptation technique. Then, the network depth and width are dynamically adjusted in the Optuna optimization framework to accommodate more complex fault types in the target domain. Finally, the performance of the model is further improved by automatically optimizing the hyperparameters. The experimental results show that the model exhibits high accuracy in the diagnosis of different fault types, especially in the face of complex and variable industrial environments, demonstrating strong adaptability and robustness. The method provides an effective solution for fault diagnosis of intelligent devices.

## 1. Introduction

Under the development trend of intelligent and highly reliable industrial equipment, bearing fault diagnosis technology has become a key link to ensure the stable operation of modern manufacturing systems. As the core supporting components of rotating machinery, minor defects in bearings are very likely to cause chain failures or even major safety accidents under high-speed and heavy-duty working conditions, and according to statistics, about 45% of mechanical failures in industrial scenarios can be traced back to bearing anomalies, so bearing fault diagnosis has always been a hotspot for research.

With the increasing maturity of machine learning technology, technicians have proposed a series of intelligent diagnostic methods [1,2,3], which significantly improve the rolling bearing fault diagnosis capability. Lee and Sabater proposed the conversion of time series data into visibility graphs from which persistent homology is applied to extract topological features based on topological data analysis (TDA) [4]. Shinde et al. proposed the basic bearing frequencies withdrawn from the vibration response as novel extracted features and provided the data to a K nearest neighbor network (KNN) for fault classification [5]. Phan et al. proposed to apply EMD and EEMD decomposition on the signals and then combine the features extracted from the preprocessed and decomposed signals to select the information function through BGWO and input the selected function into the classifier, and experiments proved that this method can lead to a significant improvement in ML classification accuracy [6]. However, ML still faces some problems when used for bearing fault diagnosis: the performance of the model relies on a large amount of labeled data, and existing methods may suffer from insufficient generalization ability when dealing with complex conditions or new fault types.

In recent years, deep learning has been introduced into the field of bearing fault diagnosis, and a series of results have been achieved [7,8,9]. Shan et al. proposed a Mel-CNN model [10] for the fault diagnosis of noisy motor bearings and achieved more accurate and reliable diagnostic results compared with the previous model. The Mel-CNN model utilizes variational mode decomposition (VMD) to remove the high-frequency components of the motor noise, extracts Mel spectral acoustic features, and re-extracts the Mel acoustic features with the help of convolutional neural network (CNN). Li et al. proposed to find the optimal hyperparameter combination of long-short-term memory (LSTM) network based on SOA [11]. Zhou et al. proposed a fault diagnosis framework based on evidence theory and the improved visual set group (VGG) neural network (EVGG) to obtain accurate and reliable diagnostic results with additional estimation of the prediction uncertainty to obtain accurate and reliable diagnostic results [12]. Chen and Zou [13] trained separate convolutional neural network (CNN) and long short-term memory network (LSTM) models and implemented three different fusion strategies. However, the application of deep learning in the field of bearing fault diagnosis still faces many challenges [14,15,16]: the performance of deep learning models is highly dependent on hyperparameters, which requires professional technicians to spend a lot of time on debugging; the generalization ability of deep learning models is limited; and the deep learning models require that the test data be distributed in the same way as the training data.

In order to lower the threshold of machine learning [17] and shorten the model development cycle, automated machine learning (AutoML) has become a hot research topic in recent years [18,19,20]. There are already many mature AutoML platforms and tools, like Google AutoML, Auto-sklearn, H2O.ai, TPOT, etc. A number of new tools are being developed as research progresses. Feurer et al. proposed Auto-sklearn 2.0 [21], an automatic learning system based on Auto-sklearn 1.0, which enables AutoML to handle large datasets well by using a new, meta-featureless meta-learning technique with a successful bandit strategy for budget allocation. Neutatz et al. proposed Caml [22], a model that uses meta-learning to automatically adjust its AutoML parameters to accomplish the appropriate task. However, AutoML faces many challenges: AutoML for Neural Architecture Search (NAS) and hyperparameter optimization usually requires significant computational resources [23], especially when dealing with complex deep learning tasks [24]; when the dataset is not large enough or the features are not well chosen [25], it is difficult for AutoML to guarantee the model’s ability to generalize to new datasets [26]; and AutoML is susceptible to local optimal solutions in the optimization process.

With the development of transfer learning [27,28,29,30], there is hope that the problems associated with deep learning can be solved. Transfer learning shortens the model development cycle and improves the learning efficiency and performance of new tasks by transferring relevant knowledge from previously trained tasks to related tasks. Wu et al. proposed the use of joint distribution adaptation to reduce the difference in probability distributions between the auxiliary dataset and the target domain dataset and introduced the GWO algorithm for adaptive learning of key parameters of the model [31]. Huo et al. proposed an enhanced transfer learning method based on a linear superposition network [32] for rolling bearing fault diagnosis, which improves the structure of a one-dimensional convolutional neural network (1D-CNN) by constructing linear superposition of convolutional blocks, which enhances the ability of the model to extract fault features. Thuan proposed a robust transfer learning based on a multi-layer maximum mean difference loss function [33], and the model achieved better results in cross-machine scenarios. Zhang et al. proposed a fault diagnosis method based on an image information fusion and visual transform (ViT) transfer learning model [34]. Despite the good results of transfer learning in the field of bearing fault diagnosis, there are still some issues to be aware of: existing public bearing fault datasets are not as large as datasets such as ImageNet, The Pile, etc., which are small and medium-sized datasets; however, transfer learning cannot provide ideal diagnosis with existing models and learning strategies in cases where the source domain dataset is not very large or the source domain dataset is smaller than the target domain dataset.

The above issues are summarized below:(1)There are no publicly available large bearing failure datasets that can be used to train neural networks, leading to poor performance of transfer learning in solving multi-failure complex problems.(2)Deep learning to find the optimal hyperparameters for a model takes a lot of developer time and effort.(3)Existing AutoML platforms and tools generally require significant computational resources when performing Neural Architecture Search (NAS).

In this paper, we propose an automated transfer learning bearing fault diagnosis model for multi-fault complexity. First, transfer learning is utilized to apply a well-trained model in the source domain ($D_{s}$) to the task in the target domain ($D_{t}$), and the distribution difference between the source and target domains is reduced by domain adaptation. Then, a neural network depth and width dynamic adjustment module is embedded in Optuna [35], which not only dynamically freezes and unfreezes the layers of the neural network but also chooses whether to increase and how to increase the depth and width of the neural network within a certain range to adapt to larger data sizes and more complex fault types in the target domain through Optuna’s built-in optimization algorithm. Finally, model hyperparameters including learning rate, number of training rounds, task loss weights, and domain loss weights are optimized by Optuna to further improve the model performance. The innovation of the proposed method is that Optuna automatically optimizes the depth and width of the neural network to adapt to the complexity of the target domain while using ADA to reduce the difference in the distribution of the source and target domains. This approach not only improves the generalization ability of the model across different fault types and complex environments but also reduces the need for manual parameter tuning and improves the accuracy and efficiency of fault diagnosis. The contributions of this paper are summarized as follows:(1)Applying pre-trained models to the target domain using transfer learning avoids the use of AutoML for Neural Architecture Search (NAS), reduces the demand for computational resources during model development, and significantly shortens the development cycle relative to building models from scratch.(2)Compared with the fine-tuning [36] strategy commonly used in transfer learning, this paper proposes the method of embedding the neural network depth and width dynamic adjustment module, which largely improves the model’s ability to classify complex faults and effectively copes with the current lack of large public bearing fault datasets.(3)The introduction of Optuna to optimize the hyperparameters of the model no longer requires professionals to spend a lot of time on manual debugging, which lowers the threshold of model development and use and is conducive to the promotion of “intelligence” in various industries.

## 2. Theoretical Foundation

### 2.1. Adversarial Domain Adaptation

Adversarial domain adaptation (ADA) [37] and generative adversarial networks (GANs) [38] share similar adversarial ideas. The main goal of ADA is to reduce the distributional differences between the source and target domains, allowing models trained in the source domain to migrate to the target domain. When ADA is applied to transfer learning and domain adaptation tasks, the aim is for the model to achieve better adaptation between the source and target domains. The main goal of GANs is to generate realistic data samples, enabling the generator to produce samples that are similar to the real data distribution, while the discriminator’s task is to determine whether the input samples are real or not.

#### 2.1.1. Basic Idea

Adversarial domain adaptation is inspired by generative adversarial networks (GANs), and its core idea is to introduce an “adversarial” process into the training process. Specifically, it includes the following:

Feature extractor: The role of the feature extractor is to extract useful features from the input data. In ADA, the feature extractor is usually a deep neural network that is used to extract generic features of the source and target domain data.

The discriminator aims to determine whether the input features come from the source domain or the target domain by training the discriminator to help the generator learn a more generalized feature representation.

The goal of adversarial training is to obtain a domain-invariant feature representation for the generator by optimizing the game process of the generator and the discriminator, allowing the model to generalize better to the target domain.

#### 2.1.2. Processes for Adversarial Domain Adaptation

Adversarial domain adaptation typically involves the following major steps:The feature extractor extracts features from the source and target domain data, usually implemented using deep neural networks (e.g., convolutional neural networks, fully connected networks, etc.).Discriminator: the task of the discriminator is to determine whether the input features are from the source or target domain. Ideally, after training, the discriminator will not be able to distinguish between features from the source and target domains.The classifier is responsible for classification tasks based on extracted features (e.g., image classification, text classification, etc.).Adversarial training aims to optimize the generator and discriminator by backpropagation. The goal of the discriminator is to differentiate as much as possible between data in the source and target domains, while the goal of the feature extractor and classifier is to try to make the discrimination as ineffective as possible by generating an adversarial process.

#### 2.1.3. Principles and Formulas: An Example of Domain Adversarial Neural Network (DANN)

Basic Settings and Symbol Definitions

$D_{s}={\left\{\left(x_{i}^{s},y_{i}^{s}\right)\right\}}_{i=1}^{n_{s}}$ is the source domain dataset, where $x_{i}^{s}$ is the input data from which it originated and $y_{i}^{s}$ is the label of the source domain.

$D_{t}={\left\{x_{i}^{t}\right\}}_{i=1}^{n_{t}}$ is the target domain dataset, where $x_{i}^{t}$ is the input data for the target domain, which is unlabeled.

F is the feature extractor that maps the input x to the shared feature space $H:F\left(x\right)$.

C is the classifier for a feature-based $F\left(x\right)$ for a source domain classification task.

D is an adversarial discriminator that is used to determine whether the features of a sample come from the source or target domain.

2.Categorized losses

In the source domain, the goal is to train a classifier C, which maps the inputs from the source domain to the category labels y. The categorization loss is usually measured using Cross-Entropy Loss to measure the performance of the source domain classification.(1)$$L_{c}=-\frac{1}{n_{s}}\sum_{i=1}^{n_{s}}\log P\left(y_{i}^{s}|F\left(x_{i}^{s}\right)\right)$$
where $P\left(y_{i}^{s}|F\left(x_{i}^{s}\right)\right)$ is the predicted probability of classification of the classifier C on the source domain sample $x_{i}^{s}$.

In conjunction with this paper, an increase in the depth of the neural network improves the feature extraction capability and allows the model to learn more complex features, thereby improving classification accuracy and reducing classification loss. An increase in the width of the neural network improves the feature representation per layer, allowing the model to handle more features, improving classification performance, and reducing classification loss. Considering the effect of depth and width adjustments on the classification loss, a regularization term is introduced into the loss function to penalize overly complex model structures and thus avoid overfitting, and the new classification loss function is(2)$${L^{\prime }}_{c}=L_{c}+\lambda_{reg}\cdot R(L_{new},L_{lnew})$$
where $\lambda_{reg}$ is the regularization factor and $R(L_{new},L_{lnew})$ is a regularization term resulting from depth and width adjustments to limit the network complexity. This can improve the classification accuracy while avoiding the overfitting problem caused by an overly complex network.

3.Adversarial loss

The core of adversarial domain adaptation is to make features from the source and target domains as indistinguishable as possible in the shared feature space. To this end, we introduce an adversarial discriminator D, whose task is to determine whether a given feature comes from the source domain. Adversarial loss is often used to measure the performance of the discriminator D using the logarithmic loss of the discriminator:(3)$$L_{a}\left(D\right)=-\frac{1}{n_{s}}\sum_{n=1}^{n_{s}}\log D\left(F\left(x_{i}^{s}\right)\right)-\frac{1}{n_{t}}\sum_{n=1}^{n_{t}}\log\left(1-D\left(F\left(x_{i}^{t}\right)\right)\right)$$
where $D\left(F\left(x_{i}^{s}\right)\right)$ is the probability that the discriminator D predicts that the source domain sample feature $F\left(x_{i}^{s}\right)$ comes from the source domain and $D\left(F\left(x_{i}^{t}\right)\right)$ is the probability that the discriminator D predicts that the target domain sample feature $F\left(x_{i}^{t}\right)$ comes from the target domain. The discriminator D tries to maximize this loss so that it accurately distinguishes between samples from the target and source domains.

For the method proposed in this paper, the increase in the depth of the neural network helps the generator to learn richer domain invariant features, which makes the data features in the source and target domains more indistinguishable and thus effectively reduces the adversarial loss. The increase in the width of the neural network enhances the feature representation of the generator and discriminator, allowing the generator to learn more domain-invariant features, thus reducing adversarial loss. At the same time, the increase in width also enhances the discriminator’s discriminative ability, making it more difficult to distinguish between features in the source and target domains and helping to improve the model’s ability to migrate across domains. The adjusted adversarial loss function can be expressed as(4)$${L_{a}}^{\prime }=-\sum_{i=1}^{N_{s}}\log(D(x_{i};L_{new},N_{lnew}))-\sum_{i=1}^{N_{t}}\log(x_{j};L_{new},N_{lnew})$$
where $D(x_{i};L_{new},N_{lnew})$ is the output of the discriminator for either the source domain sample or the target domain, dependent on the adjusted depths $L_{new}$ and $N_{lnew}$.

4.Objectives of Confrontation Training

In adversarial training, the feature extractor and the discriminator are trained in a game. The goal is for the feature extractor to learn a domain-invariant feature representation that makes it impossible for the discriminator to distinguish between samples in the source and target domains. The loss function of the feature extractor is the negative of the adversarial loss; we want to maximize the discriminator’s error. Therefore, the function of the feature extractor is(5)$${L^{\prime }}_{a}\left(F\right)=-{L^{\prime }}_{a}\left(D\right)$$

5.Joint optimization

The training objective of DANN [39] is to optimize both classification loss and adversarial loss with a total loss function:(6)$$L_{total}={L^{\prime }}_{c}+\lambda {L^{\prime }}_{a}\left(F\right)$$
where λ is the hyperparameter that balances the classification loss and the adversarial loss. Classification loss ($L_{c}$) is used to ensure that the classification task on the source domain is learned correctly. Adversarial loss ($L_{a}$) optimizes the feature extractor and discriminator by backpropagation, making it difficult to distinguish between features of the source and target domains in the shared space. The basic DANN architecture is shown in Figure 1.

## 3. Proposed Methodology

### 3.1. Adversarial Domain Adaptive Transfer Learning

Adversarial domain adaptation (ADA) is a transfer learning method based on the idea of adversarial training in the field of deep learning. Its main purpose is to solve the problem of distributional differences between the source domain and the target domain, thus enabling the model to migrate the knowledge learned from the source domain to the target domain.

#### 3.1.1. Source Domain Model Selection and Training

Convolutional neural networks (CNNs) [40] are well suited for processing time-series signals because they are effective in extracting useful features from localized regions. The basic CNN architecture is shown in Figure 2. In bearing fault classification, a CNN can automatically extract time-frequency features in the signal without manual feature engineering. The steps for building and training the pre-trained model are as follows.

Data preprocessing

Vibration signals from bearings are usually time series data. These data need to be preprocessed appropriately before using CNNs, The data preprocessing flowchart is shown in Figure 3:

Signal denoising: the vibration signal may contain noise from the environment, so it is necessary to use a filter (low-pass filter) to remove high-frequency noise, assuming that the original signal is $x\left(t\right)$ and the filtered signal $\overset{\wedge }{x}\left(t\right)$ can be calculated by convolution with the filter $h\left(t\right)$:(7)$$\overset{\wedge }{x}\left(t\right)=x\left(t\right)*h\left(t\right)$$

Signal segmentation: the original vibration signal is segmented, and each segment is used as a sample for model training.

Feature scaling: normalization of the signal is required to ensure consistent scaling of each input feature.(8)$${\overset{\wedge }{x}}_{i}=\frac{x_{i}-\mu }{\sigma }$$
where $x_{i}$ is a sample from the input signal matrix, μ is the sample mean, σ is the standard deviation, and ${\overset{\wedge }{x}}_{i}$ is the normalized sample.

2.Building CNN models

Convolutional layer: a convolutional layer is used to extract localized features from the input data. When dealing with one-dimensional signals, the size of the convolution kernel is taken to be a smaller value, which helps in capturing the local features of the signal. In the case of one-dimensional convolution, the formula for convolution operation is(9)$$y\left(t\right)=\sum_{k=0}^{k-1}x\left(t-k\right)\cdot \omega \left(k\right)$$
where $x\left(t\right)$ is the input signal, $\omega \left(t\right)$ is the convolution kernel, and $y\left(t\right)$ is the convolved output.

Activation layer: the activation function is a ReLU (Rectified Linear Unit), which increases the nonlinear capability of the model, thus improving the classification performance, and is defined as(10)ReLUx=max0,x

Pooling layer: the pooling layer is usually used to reduce the size of the feature map while retaining important information. When dealing with 1D vibration signals, this paper uses 1D Max Pooling:(11)$$y\left(t\right)=\max\left\{x\left(t\right),x\left(t+1\right),\cdots ,x\left(t+k-1\right)\right\}$$
where k is the size of the pooling window, $x\left(t\right)$ is the input signal, and $y\left(t\right)$ is the pooled input.

Fully connected layer: in the last layers of the network, the features extracted through the convolution and pooling layers are spread into a one-dimensional vector and passed into the fully connected layer for classification decisions:(12)y=Wx+b
where x is the input vector, W is the weight matrix, b is the bias term, and y is the output.

Output layer: the output layer uses the Softmax activation function, which, for a multicategorization problem, is capable of outputting the probability of each category:(13)$$P\left(y_{i}\right)=\frac{e^{z_{i}}}{\sum_{j}e^{z^{j}}}$$
where $z_{i}$ is the score of the first i category and $P\left(y_{i}\right)$ is the probability of that category.

3.Model training

In order to train the CNN model, we need to use a labeled vibration signal dataset. The training process includes the following steps:

Define loss function: the loss function used in this paper is Cross-Entropy Loss, which is used to calculate the difference between the predicted and actual categories:(14)$$L\left(y,\overset{\wedge }{y}\right)=-\sum_{i}y_{i}\log\left(\overset{\wedge }{y_{i}}\right)$$
where $y_{i}$ is the probability distribution of the actual categories and $y_{i}$ is the probability distribution of the predicted categories.

Optimizer: the optimizer used in this paper is Adam, which can adaptively adjust the learning rate to improve the training efficiency. the updated formula of the Adam optimization algorithm is(15)$$\theta_{t}=\theta_{t-1}-\eta \cdot \frac{{\overset{\wedge }{m}}_{t}}{\sqrt{{\overset{\wedge }{v}}_{t}}+\varepsilon }$$
where $\theta_{t}$ is the parameter at step t; η is the learning rate; ${\overset{\wedge }{m}}_{t}$ and ${\overset{\wedge }{v}}_{t}$ are the first-order and second-order moment estimates of the gradient, respectively; and ε is a constant that prevents division by zero.

Training process: the training data are input into the CNN, and the weights of the model are optimized after several rounds of iterations until the loss function converges.

4.Evaluation and Testing of Models

After training the model, it needs to be evaluated. The evaluation metrics chosen in this paper are accuracy and F1 score.

#### 3.1.2. Domain Adaptive Migration of Pre-Trained Models

Developing a new model from scratch often requires a long lead time, but there already exist many models that have been designed and debugged with a lot of effort by professionals, and these models are able to perform well on the corresponding datasets. Transfer learning is able to apply the mature models to the relevant tasks. The adversarial training process is as follows:Pre-training dissemination

Input data: each batch of data in the source domain training set is fed into the model for forward propagation.

Feature extraction: the data are passed through the feature extraction part of the model to generate a shared feature representation f, and useful features are extracted from the input data:(16)$$f=\phi \left(x\right)$$
where $\phi \left(x\right)$ is the feature extractor function.

Task classification: the output of the feature extractor is passed to the task classifier for classification prediction. The task classifier calculates the classification loss and is trained on the target label.

Domain classification: the features inverted by the GradientReverseLayer are fed into the DomainClassifier for domain discrimination. The DomainClassifier tries to determine whether the features come from the source or target domain, and the task is to identify the source (source or target) of these features.

2.Calculating the loss function

Task classification loss: task classification loss is computed using CrossEntropyLoss, which measures the difference between the classifier output and the true label.

Adversarial loss: adversarial loss is computed through DomainClassifier with the goal of minimizing the distinction between source and target domain features. With the GradientReverseLayer (GRL), the feature extractor adjusts the parameters so that the domain discriminator cannot distinguish between source and target domain data.

Total loss function: the final total loss function is the weighted sum of the task classification loss and the adversarial loss.

3.Backpropagation and Optimization

Backpropagation: by calculating the gradient of task loss and adversarial loss and updating the weights of the model using the optimizer, the gradient update formula is(17)$$\theta_{t+1}=\theta_{t}-\eta \cdot \nabla_{\theta }L_{total}$$
where $\theta_{t}$ is the parameter, η is the learning rate, and $\nabla_{\theta }L_{total}$ is the gradient of the total loss function.

Optimizer: the optimizer Adam adjusts the model parameters according to the gradient of the loss function. At the end of each training round, the optimizer updates the weights of the feature extractor, classifier, and domain classifier based on the backpropagated gradient.

4.Training and validation

Training phase: in each epoch, the training data from the source domain are trained by the model and the training accuracy is calculated.

Validation phase: at the end of each epoch, the migration effectiveness of the model is evaluated using validation data from the target domain. Evaluation metrics such as accuracy, precision, recall, and F1 score are calculated to measure the performance of the model on the target domain.

5.Updating the best indicators

After each epoch, the code calculates the accuracy, precision, recall, and F1 score on the validation set and updates the best results. Eventually, the optimal accuracy, precision, recall, and F1 score are recorded to choose the best hyperparameters.

### 3.2. Structural Conditioning Module Incorporating Optuna Hyperparametric Optimization Algorithm

The module is able to dynamically adjust the depth and width of the neural network. At the same time, the module, in combination with Optuna, chooses whether to increase the depth or width of the network or to adjust these parameters within a given range by means of a built-in optimization algorithm. This dynamic tuning not only improves the performance of the model on the target domain but also optimizes computational resources to avoid overfitting or underfitting.

#### 3.2.1. Defining the Network Structure

We use the pre-trained model as a base and allow Optuna to adjust the depth, width, and freeze layer of the network during training.

#### 3.2.2. Setting up Optuna to Optimize Space

Optuna [33] selects the appropriate hyperparameter optimization space to search for the optimal depth and width. The Optuna optimization problem can be defined as(18)$$Objective\left(D,W\right)=Loss\left(D,W\right)+\alpha \cdot Complexity\left(D,W\right)$$
where $Loss\left(D,W\right)$ is the loss of the model on the validation set, α is the weight that weighs the loss against the computational complexity, and $Complexity\left(D,W\right)$ is the computational complexity associated with the network depth D and width W.

#### 3.2.3. Depth and Width Adjustment

1. Depth tuning: the model increases the depth of the network by adding convolutional layers, fully connected layers, etc. At the same time, the model will decide whether to add layers and how many layers. Assuming that the initial network has $L_{0}$ layers and Optuna has the option to add and subtract layers, the new depth of the network is(19)$$D=L_{0}+L_{add},L_{add}\in \left\{0,1,2,\cdots ,L_{add}\right\}$$
where $L_{add}$ is the number of layers added and $L_{\max}$ is the maximum amount of additions Optuna can select.

2. Width adjustment: for each layer, an increase in width usually means adding more neurons or a larger size convolution kernel. Optuna will determine the width of each layer based on the complexity of the data in the target domain. Assuming that the width of each layer can be expressed as $\omega_{i}$, the width is adjusted for the i th layer:(20)$$W_{i}=\omega_{i}\times \Delta W,\;\Delta W\in \left[W_{\min},W_{\max}\right]$$
where ΔW is the percentage increase in width and $\left[W_{\min},W_{\max}\right]$ is the range of width adjustment.

#### 3.2.4. Freezing and Thawing Layers

1. Freezing layers: to avoid overfitting, pre-trained layers can be frozen so that their parameters are not involved in backpropagation and optimization. Assuming that the layer i is frozen, the parameters of this layer $\theta_{i}$ remain unchanged during the training process:(21)$$\frac{\partial L}{\partial \theta }=0$$

2. Unfreeze layers: newly added layers can start training. When the network is trained to a certain point, there is an option to unfreeze the pre-trained layers in order to fine-tune them on the target domain. For unfrozen layers, weights are trained. At some stage, the parameters of the unfrozen layers can be updated:(22)$$\theta_{i}^{new}=\theta_{i}^{old}-\eta \cdot \frac{\partial L}{\partial \theta_{i}}$$
where η is the learning rate.

#### 3.2.5. Training and Assessment

After each trial, the loss and accuracy are computed through the training set, and the performance of the model on the target domain is evaluated through the validation set, which ultimately returns the evaluation metrics for the objective function, which is(23)$$F=Loss\left(D,W\right)+Accuracy\left(D,W\right)$$

### 3.3. Hyperparameter Optimization of the Model

In this paper, we use Optuna to optimize the hyperparameters of the model. With Bayesian optimization, dynamic spatial tuning of hyperparameters, and multi-objective optimization, Optuna is able to efficiently balance exploration and exploitation in the hyperparameter optimization process. New areas are explored to discover potentially optimal hyperparameters, while known good areas are exploited to progressively optimize the performance of the model. The hyperparameters being optimized include learning rate, task weight, and domain weight, and the workflow of Optuna can be divided into the following steps.

#### 3.3.1. Defining the Objective Function

The objective function is the core of the Optuna optimization process and represents the task we want to optimize. Inside the objective function, the training process of the model uses these hyperparameters and returns evaluation metrics (accuracy and F1 score), expressed in mathematical formulas as follows:(24)maxf(θ)=arg(Precision(θ),Recall(θ),F1(θ),Accuracy(θ))
where θ is the model hyperparameters, Precision(θ) is the precision, Recall(θ) is the recall probability, F1(θ) is the F1 score, and Accuracy(θ) is the accuracy.

#### 3.3.2. Defining a Hyperparametric Search Space

In the objective function, we define the search space of hyperparameters such that these hyperparameters will be explored by Optuna during each trial. The search space of the proposed method is detailed in Table 1.

#### 3.3.3. Creating an Optuna Study

Optuna uses the study object to manage the optimization process. A study is a container for Optuna that contains multiple trials, each representing a different combination of hyperparameters at a time. A study automatically manages the optimization process and selects the next set of hyperparameters based on the results of the trials.

#### 3.3.4. Implementing the Optimization Process

During the execution of the optimization, Optuna automatically selects hyperparameters based on the settings of the objective function, runs the objective function, and records the results of each trial. Each trial generates a new set of hyperparameter configurations and performs model training using these hyperparameters.

#### 3.3.5. Recording and Selecting Optimal Hyperparameters

At the end of all trials, the optimal hyperparameters and corresponding results can be viewed. The flowchart of the proposed method is shown in Figure 4.

## 4. Experimental Verification

### 4.1. Description of the Dataset

In order to verify the effectiveness of the proposed method for classifying complex faults, the CWRU bearing dataset was selected as the source domain data and the Paderborn University bearing dataset [41] was selected for the target domain data. The feature pairs of the two datasets are shown in Table 2. The advantages of this are as follows.

(1)Adaptation of cross-domain learning: the CWRU dataset as the source domain has better standardization and consistency and can be used to train a base model, whereas the Paderborn dataset as the target domain is more challenging and complex, which is relevant to the point of this paper.(2)Adaptability to real-world environments: through cross-domain learning, the model is able to learn some generalized feature extraction methods from the concise and standard CWRU dataset and then migrate to the noisy and data-complex Paderborn dataset, which is able to better cope with the variations and disturbances of real-world industrial environments.(3)Improve robustness: By training the base model on the source domain and then migrating it to the target domain for tuning, the robustness of the model in the face of complex environments, noise, and disturbances in real applications can be enhanced.

**Table 2 sensors-25-01851-t002:** Comparison of CWRU dataset and Paderborn dataset characteristics.

Feature	CWRU Dataset	Paderborn Dataset
Fault Type	Single faults	Compound faults
Fault Creation	Artificial EDM machining	Artificial machining + natural wear
Operating Conditions	Fixed load	Dynamic loads, variable speed operation
Signal Types	Vibration signals	Vibration + AE + current signals
Data Scale	Small	Large
Authenticity	Lab environment	Industrial-like scenarios
Application	Basic algorithm validation	Testing in complex scenarios

#### 4.1.1. Introduction to CWRU Bearing Dataset

The CWRU [42] dataset was provided by Case Western Reserve University and is widely used for fault diagnosis, machine learning, and health monitoring studies. The dataset contains bearing vibration data under several different operating conditions, which is used to study different failure modes of rolling bearings, and the signal acquisition device is shown in Figure 5. The dataset is characterized as follows.

(1)The failure types include inner ring failure, outer ring failure, rolling body failure, and mixed failure (inner ring and outer ring failure at the same time). This paper selects the inner ring failure and outer ring failure data for research.(2)Experimental setup: Datasets were acquired at a fixed load and multiple rotational speeds with four load settings (loaded to 0, 1/3, 2/3, and 1× the rated load) and different rotational speeds (1797 rpm, 1750 rpm, 2100 rpm, etc.).(3)The signal type mainly uses vibration signals; the acquisition directions for the X-axis, Y-axis, and Z-axis; and a sampling frequency of 12 kHz. The data acquisition environment is relatively simple, with less interference.

**Figure 5 sensors-25-01851-f005:**
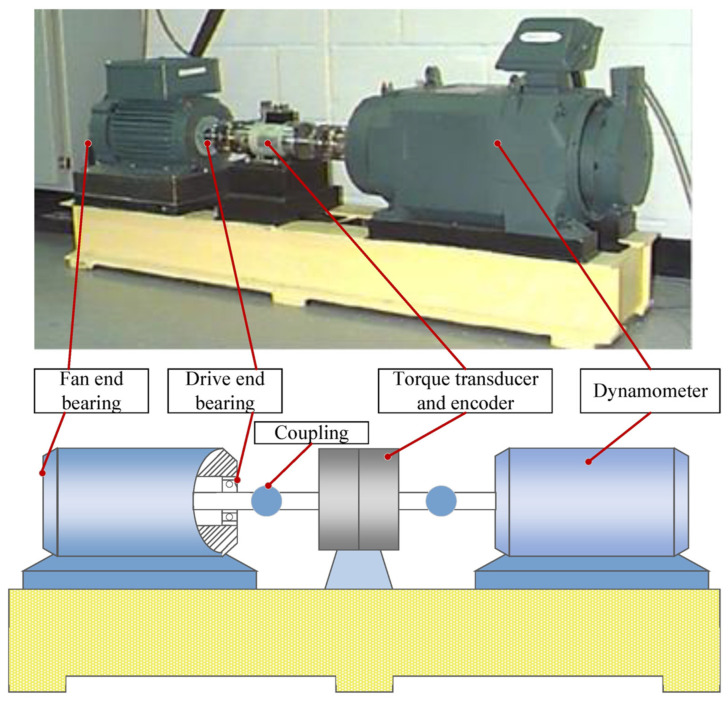
CWRU bearing dataset test bed.

#### 4.1.2. Introduction to Paderborn University Bearing Dataset

The Paderborn dataset was provided by the University of Paderborn, Germany, and its data acquisition device is shown in Figure 6. The dataset focuses on the simulation of bearing failure data in industrial environments, aiming to study how to diagnose failures in more complex and variable environments, and only the vibration signals in this dataset are studied in this paper. The dataset is characterized as follows.

(1)The failure types include inner ring failures, outer ring failures, and mixed inner and outer ring failures, and the failure types in the dataset are a mixture of manual processing and natural aging failures. These fault types are selected in the experiment.(2)Experimental setting: This dataset uses dynamic load and variable speed operating conditions to simulate load variations closer to those found in industrial environments. The vibration signals in the experiments are collected by multiple sensors, including acoustic emission (AE) and current signals in addition to accelerometers.(3)The signal type contains vibration signals (X- and Y-axis), acoustic emission signals (AE), and current signals, with sampling frequencies typically 25.6 kHz or higher. In this paper, only vibration signals are studied.

**Figure 6 sensors-25-01851-f006:**
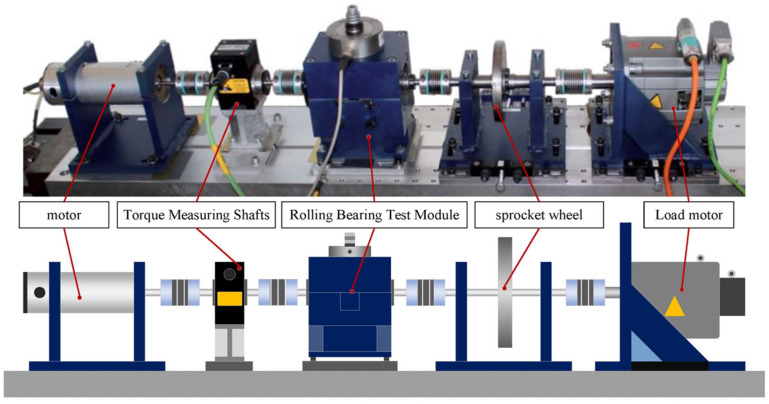
Paderborn University bearing dataset test bed.

### 4.2. Experimental Setup

In order to demonstrate the effectiveness of the proposed method, this paper sets up seven sets of comparison experiments:(1)In order to show that the proposed method occupies less computational resources than the AutoML tool alone, the proposed method is trained with AutoGluon and Auto-Keras under the same conditions, and its running time on the GPU is compared;(2)To illustrate the need for model migration and hyperparameter optimization, the proposed method is compared with RNN and CNN;(3)In order to demonstrate that the addition of a neural network depth and width dynamic adjustment module can facilitate the positive migration of pre-trained models and the ability to classify complex faults, the proposed method is compared with DANN, MMD, and DAN.

In order to make the experimental results more convincing, 10 repetitions of each model were conducted, and the main parameters of TL and DL were set consistently: the learning rate was 0.001, the batch size was 128, and the epoch was 20, as detailed in Table 3.

### 4.3. Analysis of Experimental Results

In order to clearly demonstrate the model performance, accuracy (Table 4), F1 score (Figure 7), confusion matrix (Figure 8), and GPU time (Figure 9) are selected to compare the performance of each method on the target domain dataset.

In the confusion matrix, artificial damage in the outer ring corresponds to label 0, true damage in the outer ring corresponds to label 1, artificial damage in the inner ring corresponds to label 2, true damage in the inner ring corresponds to label 3, damage in the outer and inner rings corresponds to label 4, and no damage corresponds to label 5. The horizontal coordinate is the predicted label, and the vertical coordinate is the true label.

Analyzing the experimental data, the following conclusions can be obtained:(1)Comparing method 2 and method 3, the accuracy of the proposed method is better than that of AutoGluon and Auto-Keras, which indicates that the proposed method is able to automatically optimize the hyperparameters of the model while guaranteeing higher fault diagnosis accuracy than the more advanced AutoML tools. This is due to the fact that the proposed method uses TL combined with the structural adjustment module to construct the model in the case where the target domain task is related to the source domain task, which is more advantageous than the model construction methods of AutoGluon and Auto-Keras.(2)According to Figure 9, it can be seen that the GPU time of the proposed method is less than those of AutoGluon and Auto-Keras. This in part reflects the fact that the proposed method requires less computational resources than AutoGluon and Auto-Keras. This is due to the fact that TL applies pre-trained models to the target domain, which accurately reduces the search space and thus accelerates the NAS process.(3)Comparing method 4 and method 7, it can be seen that the diagnostic accuracy of DANN is slightly higher than that of CNN, indicating that adversarial domain adaptive migration learning can effectively improve the model’s ability to diagnose across domains, and the use of this strategy can effectively promote positive migration.(4)Comparing method 1, method 4, method 5. and method 6 shows that the classification accuracy and F1 score of the proposed method are better than those of TL. This indicates that the structural adjustment module can effectively improve the adaptive ability of the pre-trained model when facing a complex fault dataset.

## 5. Conclusions

In this paper, a domain adversarial transfer learning bearing fault diagnosis model incorporating structural adjustment modules is proposed. First, based on the transfer learning framework, the model that has been trained and matured in the source domain is migrated to the target domain task, and the data distribution difference between the source and target domains is narrowed by the adversarial domain adaptation technique to enhance the cross-domain generalization ability of the model. Then, the Optuna optimization framework is embedded with a module to dynamically adjust the depth and width of the neural network, which achieves the dual function of flexibly freezing (locking parameters) or unfreezing (enabling training) specific network layers according to the requirements of the task in the target domain, and at the same time, utilizing the built-in optimization algorithms (e.g., Bayesian optimization) in Optuna to automatically decide whether to expand the depth of the network (increasing the number of layers) or the width of the network (increasing the number of neurons), determine the expansion range (increase the number of neurons), and determine the scope of expansion to accommodate larger data sizes and more complex failure modes in the target domain. Finally, key hyperparameters of the model are globally optimized by Optuna, including the learning rate, the number of training iterations, the balance coefficient between the task classification loss weight and the domain adaptation loss weight, etc., so as to further improve the diagnostic accuracy and robustness of the model on the target domain.

However, the approach proposed in this paper may have challenges when facing situations where the source and target domains differ significantly, especially when the target domain contains new fault types that are not present in the source domain. Although the distributional differences between domains can be somewhat reduced by adversarial domain adaptation (ADA), it may be difficult for this approach to fully cope with new failure modes in the target domain. For such completely new fault types, the model may need more target domain data or additional policies to enhance its adaptability to the new fault types. In addition, although the dynamic adjustment of depth and width can improve the performance of the model, the adjusted model may still face the problem of insufficient generalization ability if the difference between the target domain and the source domain is too large. Therefore, a combination of more domain-specific knowledge or self-supervised learning methods will be explored in the future to cope with this limitation.

## Figures and Tables

**Figure 1 sensors-25-01851-f001:**
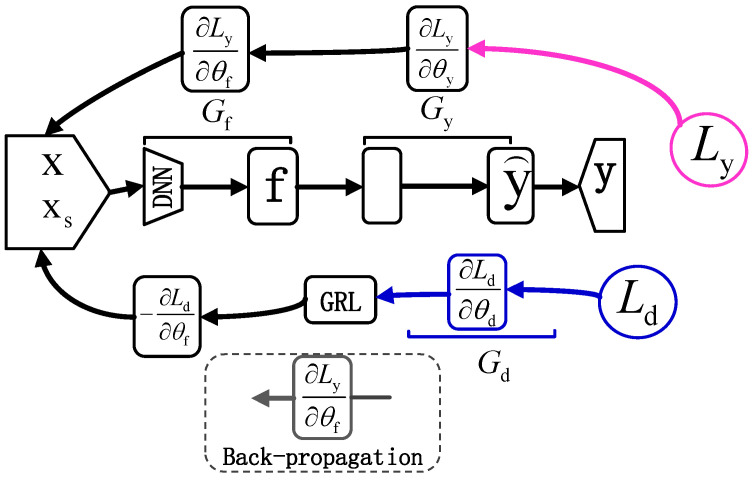
DANN basic architecture.

**Figure 2 sensors-25-01851-f002:**
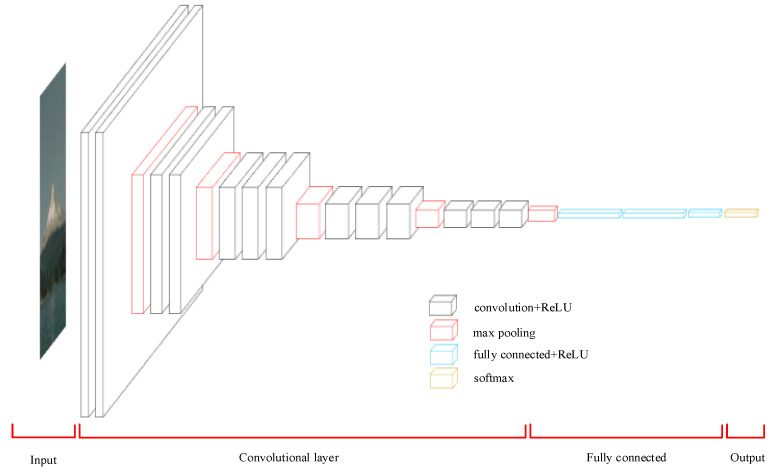
CNN architecture diagram.

**Figure 3 sensors-25-01851-f003:**
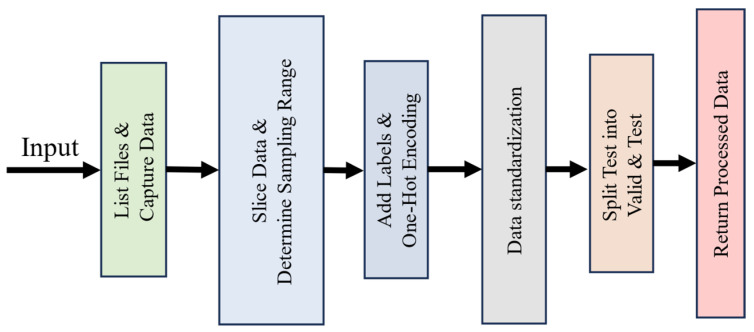
Data preprocessing.

**Figure 4 sensors-25-01851-f004:**
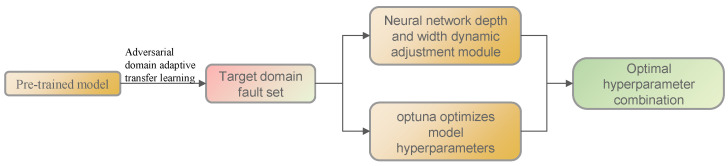
Flowchart of the proposed method.

**Figure 7 sensors-25-01851-f007:**
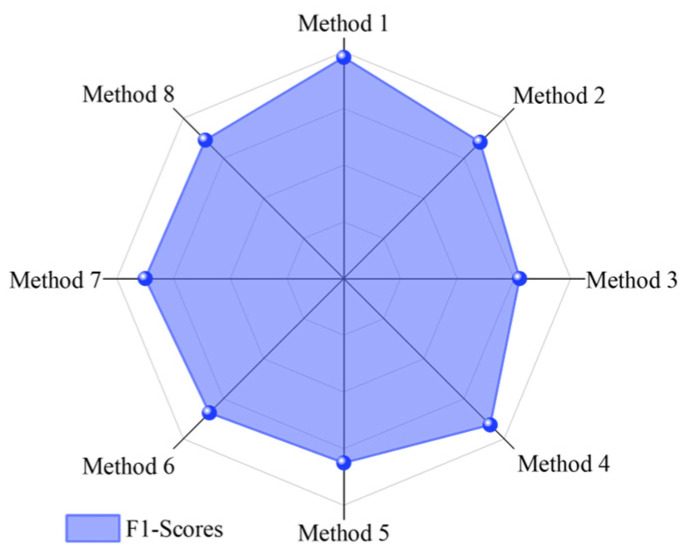
F1 scores for the eight models.

**Figure 8 sensors-25-01851-f008:**
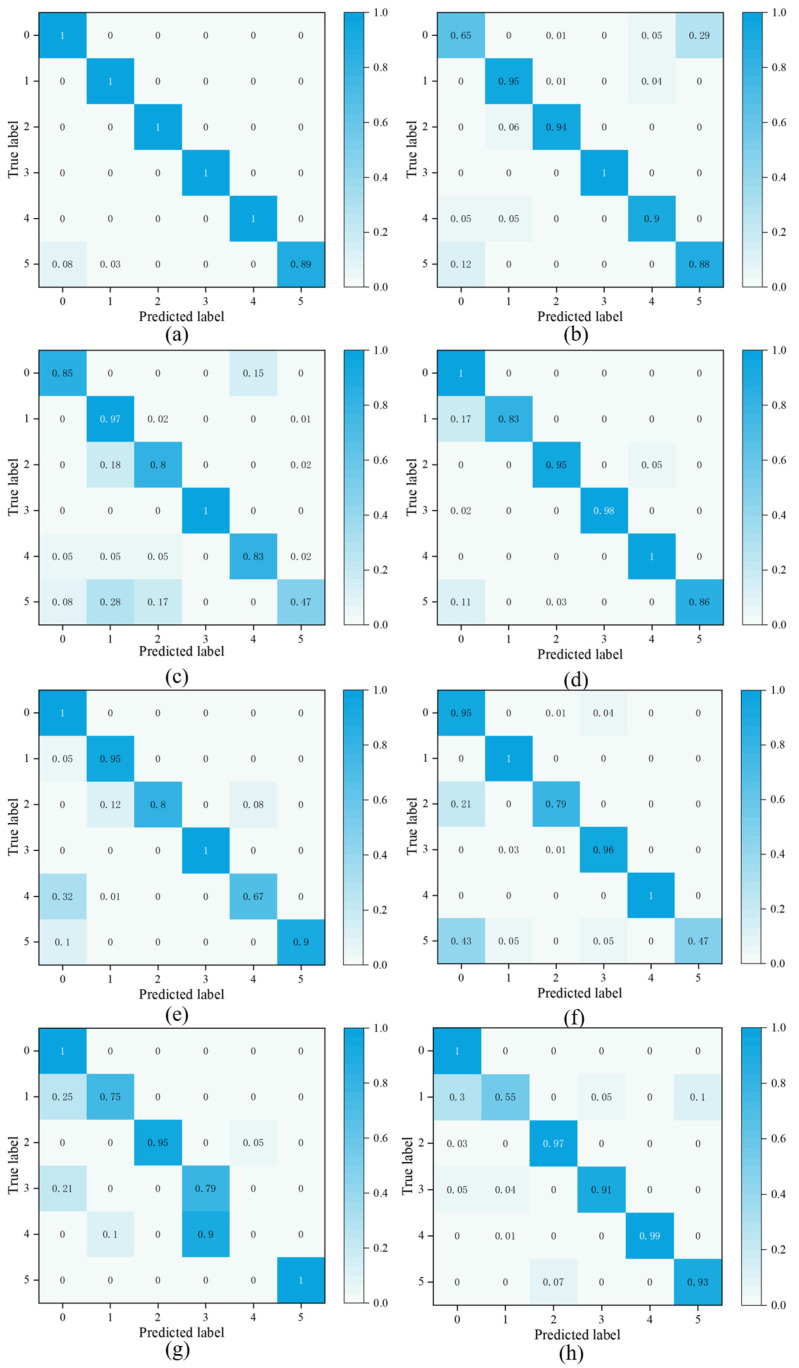
Confusion matrix for eight models. (**a**) Method 1. (**b**) Method 2. (**c**) Method 3. (**d**) Method 4. (**e**) Method 5. (**f**) Method 6. (**g**) Method 7. (**h**) Method 8.

**Figure 9 sensors-25-01851-f009:**
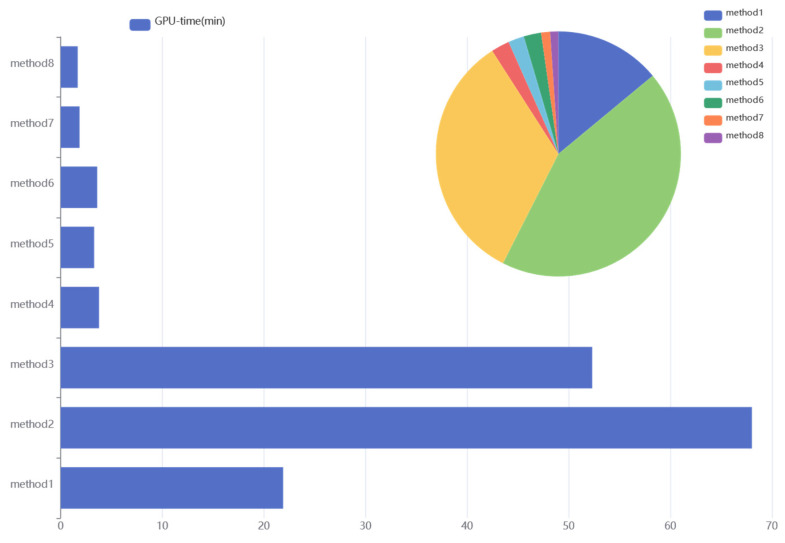
GPU time.

**Table 1 sensors-25-01851-t001:** Hyperparametric search space.

Convolutional Layer Count	Convolutional Layer Output Channel Number	Convolution Kernel Size	Convolutional Stride	Full Connection Layer Quantity	Fully Connected Layer Neuron Count	Learning Rate	Task Weight	Domain Weight
2–10	16–256	2–64	1–16	1–3	16–256	1 × e^−5^–1 × e^−2^	0–2	0–1

**Table 3 sensors-25-01851-t003:** Main parameters of the eight models.

No.	Method	Learning Rate	Batch Size	Epoch	Trial	Hard Ware
1	Proposed Method		128	20	150	GPU
2	AutoGluon		128	20	150	GPU
3	Auto-keras		128	20	150	GPU
4	DANN	0.001	128	20		GPU
5	MMD	0.001	128	20		GPU
6	DAN	0.001	128	20		GPU
7	CNN	0.001	128	20		GPU
8	RNN	0.001	128	20		GPU

**Table 4 sensors-25-01851-t004:** Accuracy of the eight models.

No.	Method	Accuracy (%)
1	Proposed Method	98.53 ± 0.35
2	AutoGluon	88.75 ± 0.22
3	Auto-keras	81.25 ± 0.13
4	DANN	92.47 ± 5.10
5	MMD	85.46 ± 4.05
6	DAN	86.24 ± 1.60
7	CNN	89.56 ± 1.45
8	RNN	89.23 ± 1.32

## Data Availability

The experimental data can be downloaded from https://engineering.case.edu/bearingdatacenter/apparatus-and-procedures (accessed on 13 March 2025) and https://mb.uni-paderborn.de/kat/forschung/kat-datacenter/bearing-datacenter (accessed on 13 March 2025).

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
