# Peer review of "Domain Adversarial Transfer Learning Bearing Fault Diagnosis Model Incorporating Structural Adjustment Modules"

_sensors, 2025, doi:10.3390/s25061851_

Round 1

Reviewer 1 Report

Comments and Suggestions for Authors
  1. The paper needs to explicitly state the novelty of combining adversarial domain adaptation with structural adjustment modules using Optuna.
  2. The section on adversarial domain adaptation (Section 2.1) provides some formulas and steps, but the explanation feels rushed. For instance, the Domain-Adversarial Neural Network (DANN) is mentioned, but the integration of the structural adjustment module isn't thoroughly explained. How does dynamically adjusting the network depth and width affect the adversarial training process? This needs more detailed mathematical and theoretical backing.
  3. In experiments, the paper compares the proposed method with TPOT, CNN, and DANN. However, there is no mention of other state-of-the-art methods in transfer learning or AutoML. Including comparisons with more recent or competitive models would strengthen the validation.
  4. The discussion of results should be more detailed. For example, why does the proposed method perform better than DANN? Is it solely due to the structural adjustments, or are there other factors?
  5. The paper claims that the proposed method requires fewer computational resources than TPOT but doesn’t provide concrete data. Metrics like training time, memory usage, or computational cost (e.g., FLOPs) should be included to support this claim. Without quantitative evidence, the efficiency claim remains unsupported.
  6. The use of Optuna for hyperparameter optimization is mentioned, but details are sparse. What was the search space for hyperparameters? How many trials were conducted? How was the balance between exploration and exploitation managed in Optuna? Providing specifics would help readers understand the robustness of the optimization process.
  7. The paper doesn’t discuss limitations. For instance, how does the method perform if the source and target domains are vastly different? What if the target domain has entirely new fault types not present in the source domain? Addressing these scenarios would provide a clearer picture of the method's applicability.

Author Response

Comments 1: The paper needs to explicitly state the novelty of combining adversarial domain adaptation with structural adjustment modules using Optuna.

Response 1:Thank you for pointing this out. We agree with this comment. Therefore, we point out in the Introduction section the innovation of using Optuna in combination with the Adversarial Domain Adaptation and Structural Adjustment modules. This revision is at the bottom of the third page of the paper and it has been marked in green.

Comments 2: The section on adversarial domain adaptation (Section 2.1) provides some formulas and steps, but the explanation feels rushed. For instance, the Domain-Adversarial Neural Network (DANN) is mentioned, but the integration of the structural adjustment module isn't thoroughly explained. How does dynamically adjusting the network depth and width affect the adversarial training process? This needs more detailed mathematical and theoretical backing.

Response 2: Thank you for pointing this out. We agree with this comment. Therefore, in order to provide a more thorough explanation of how the structural adjustment module is integrated into adversarial domain adaptive transfer learning, we give a corresponding explanation in 2.1.3. This revision is on pages five and six of the paper and it has been marked in green.

Comments 3: In experiments, the paper compares the proposed method with TPOT, CNN, and DANN. However, there is no mention of other state-of-the-art methods in transfer learning or AutoML. Including comparisons with more recent or competitive models would strengthen the validation.

Response 3: Thank you for pointing this out. We agree with this comment. Therefore, in order to strengthen the validation effect, we added some competitive models: AutoGluon, Auto-keras, MMD, DAN, RNN. This revision is on page 15, 4.2, of the paper, and it has been marked in green.

Comments 4: The discussion of results should be more detailed. For example, why does the proposed method perform better than DANN? Is it solely due to the structural adjustments, or are there other factors?

Response 4: Thank you for pointing this out. We agree with this comment. Therefore, in response to your suggestion, we have discussed the experimental results in more detail. This revision is on page 18 of the paper and it has been marked in green.

Comments 5: The paper claims that the proposed method requires fewer computational resources than TPOT but doesn’t provide concrete data. Metrics like training time, memory usage, or computational cost (e.g., FLOPs) should be included to support this claim. Without quantitative evidence, the efficiency claim remains unsupported.

Response 5: Thank you for pointing this out. We agree with this comment. In order to compare the computational resource requirements of different models, we have reviewed a lot of information and tried many methods. The difficulties in solving this problem are that AutoML is able to utilize the built-in algorithm to automatically perform Neural Architecture Search (NAS), resulting in our inability to compute its FLOPs, and that TPOT, mentioned in the paper, defaults to training the models on the CPU, while the other models are trained on the GPU. In order to reflect the demand of each model on computational resources as truly as possible, we decided not to use TPOT in our comparison experiments, so that each model was trained on GPU and some parameters were fixed, so that by comparing the running time of the different models on GPUs we were able to reflect their demand on computational resources. The comparison effect is shown in Figure 9 on page 18 of the paper.

Comments 6: The use of Optuna for hyperparameter optimization is mentioned, but details are sparse. What was the search space for hyperparameters? How many trials were conducted? How was the balance between exploration and exploitation managed in Optuna? Providing specifics would help readers understand the robustness of the optimization process.

Response 6: Thank you for pointing this out. We agree with this comment. Therefore, we give details of the hyperparameter search space, which are given in Table 1 on page 13 of the paper. Meanwhile, we explain in 3.3 how optuna strikes a balance between exploration and exploitation, and give a multi-objective planning function, the details of which are on page 12 of the paper, where it has been marked in green.

Comments 7: The paper doesn’t discuss limitations. For instance, how does the method perform if the source and target domains are vastly different? What if the target domain has entirely new fault types not present in the source domain? Addressing these scenarios would provide a clearer picture of the method's applicability.

Response 7: Thank you for pointing this out. We agree with this comment. In response to your suggestion, we have discussed the limitations in the Conclusions section of the paper. The specifics are on page 19 of the paper and it has been marked in green.

Reviewer 2 Report

Comments and Suggestions for Authors
  1. In the abstract, please provide the background introduction to Also please address the application scenario of proposed model.
  2. In the introduction, please provide more details and examples that addressing the bearing fault diagnosis with ML methods.
  3. I didn’t see the necessity to mention AUTOML after transfer learning in the introduction. If you wanna use TL in this work, it will be more logic to mention the AutoML before TL.
  4. What is the key difference between GAN and Adversarial Domain Adaptation? My understanding is they share similar adversarial idea, but according to what is provided, I cannot differentiate them.
  5. It will be beneficial to have a example plot or diagram to show the signal preprocessing operation.
  6. There’s a typo in equation 7, 13,15,17…. Please also check others.
  7. Please provide the training and testing details, such as how many iterations,CPU?
  8. It will be more informative to add more complete parameter in the Flowchart of the proposed method

Author Response

Comments 1: In the abstract, please provide the background introduction to Also please address the application scenario of proposed model.

Response 1:Thank you for pointing this out. We agree with this comment. In response to your suggestion, we have adapted the abstract to provide a background introduction at the beginning of the abstract, and the end of the abstract describes the application scenarios of the proposed model. The details are in the abstract section on page 1 of the paper. It has been marked in purple.

Comments 2: In the introduction, please provide more details and examples that addressing the bearing fault diagnosis with ML methods.

Response 2:Thank you for pointing this out. We agree with this comment. In response to your suggestion, we have provided more examples of bearing troubleshooting using ML methods in the introduction. The specifics are in the second paragraph of the introduction, on pages 1 and 2 of the paper, and have been marked in purple.

Comments 3: I didn’t see the necessity to mention AUTOML after transfer learning in the introduction. If you wanna use TL in this work, it will be more logic to mention the AutoML before TL.

Response 3:Thank you for pointing this out. We agree with this comment. In response to your suggestion, we switched the placement of AutoML and TL in Introduction to make it more logical. The specifics are in paragraphs 4 and 5 of the Introduction section of the paper, on pages 2 and 3 of the paper. It has been marked in purple.

Comments 4: What is the key difference between GAN and Adversarial Domain Adaptation? My understanding is they share similar adversarial idea, but according to what is provided, I cannot differentiate them.

Response 4:Thank you for pointing this out. We agree with this comment. As you said, GAN and ADA have similar adversarial ideas, GAN consists of a generator and a discriminator, ADA usually contains a feature extractor and a discriminator. We have modified the paper accordingly, as described in 2.1 on page 4 of the paper. It has been marked in purple.

Comments 5: It will be beneficial to have a example plot or diagram to show the signal preprocessing operation.

Response 5:Thank you for pointing this out. We agree with this comment. This specific diagram is Figure 3 on page 8 of the paper.

Comments 6: There’s a typo in equation 7, 13,15,17…. Please also check others.

Response 6: Sorry, after speaking with the assistant editor, we believe the problem may be due to the version of the software used to edit the equations. As a solution, we will submit both the WORD and PDF files of the paper and ask the Assistant Editor to correct them before sending you a corrected version for your review.

Comments 7: Please provide the training and testing details, such as how many iterations,CPU?

Response 7:Thank you for pointing this out. We agree with this comment. We give the hyperparameter search space of the proposed method in Table 1, which is on page 13 of the paper. We give the main parameters of the proposed method and the comparison experiments in Table 3, which is on page 16 of the paper.

Comments 8: It will be more informative to add more complete parameter in the Flowchart of the proposed method.

Response 8:Thank you for pointing this out. We agree with this comment. In response to your suggestion, we have shown the complete, detailed parameters in separate Tables 1 and 3. Table 1 is on page 13 of the paper and Table 3 is on page 16 of the paper.

Round 2

Reviewer 1 Report

Comments and Suggestions for Authors

After carefully reviewing the authors' responses to my comments and examining the revised manuscript, I confirm that the authors have satisfactorily addressed the concerns raised in my review. The revisions have enhanced the clarity and rigor of the research, making significant improvements to the paper's overall quality. 

Reviewer 2 Report

Comments and Suggestions for Authors

All the problems have been appropriately addressed.